# Introducing the Microbes and Social Equity Working Group: Considering the Microbial Components of Social, Environmental, and Health Justice

Suzanne L. Ishaq,[a] Francisco J. Parada,[b] Patricia G. Wolf,[c] Carla Y. Bonilla,[d] Megan A. Carney,[e] Amber Benezra,[f] Emily Wissel,[g] Michael Friedman,[h] Kristen M. DeAngelis,[i] Jake M. Robinson,[j] Ashkaan K. Fahimipour,[k,l] Melissa B. Manus,[m] Laura Grieneisen,[n] Leslie G. Dietz,[o] Ashish Pathak,[p] Ashvini Chauhan,[p] Sahana Kuthyar,[q] Justin D. Stewart,[r] Mauna R. Dasari,[s] Emily Nonnamaker,[s] Mallory Choudoir,[i] Patrick F. Horve,[o] Naupaka B. Zimmerman,[t] Ariangela J. Kozik,[u] Katherine Weatherford Darling,[v,w] Adriana L. Romero-Olivares,[x] Janani Hariharan,[y] Nicole Farmer,[z] Katherine A. Maki,[z] Jackie L. Collier,[aa] Kieran C. O'Doherty,[bb] Jeffrey Letourneau,[cc] Jeff Kline,[dd] Peter L. Moses,[ee,ff] Nicolae Morar[gg,hh]

[a]University of Maine, School of Food and Agriculture, Orono, Maine, USA

[b]Centro de Estudios en Neurociencia Humana y Neuropsicología, Facultad de Psicología, Universidad Diego Portales, Santiago, Chile

[c]Institute for Health Research and Policy, University of Illinois at Chicago, Chicago, Illinois, USA

[d]Gonzaga University, Department of Biology, Spokane, Washington, USA

[e]University of Arizona, School of Anthropology, Tucson, Arizona, USA

[f]Stevens Institute of Technology, Science and Technology Studies, Hoboken, New Jersey, USA

[g]Emory University, Atlanta, Georgia, USA

[h]American International College of Arts and Sciences of Antigua, Antigua, Antigua and Barbuda, West Indies

[i]Department of Microbiology, University of Massachusetts, Amherst, Massachusetts, USA

[j]University of Sheffield, Department of Landscape Architecture, Sheffield, United Kingdom

[k]Institute of Marine Sciences, University of California, Santa Cruz, Santa Cruz, California, USA

[l]National Oceanic and Atmospheric Administration, Southwest Fisheries Science Center, Santa Cruz, California, USA

[m]Department of Anthropology, Northwestern University, Evanston, Illinois, USA

[n]Department of Genetics, Cell, and Development, University of Minnesota, Minneapolis, Minnesota, USA

[o]University of Oregon, Biology and the Built Environment Center, Eugene, Oregon, USA

[p]School of the Environment, Florida Agricultural and Mechanical University, Tallahassee, Florida, USA

[q]Division of Biological Sciences, University of California San Diego, La Jolla, California, USA

[r]Department of Ecological Science, Faculty of Earth and Life Sciences, Vrije Universiteit Amsterdam, Amsterdam, The Netherlands

[s]Department of Biological Sciences, University of Notre Dame, Notre Dame, Indiana, USA

[t]University of San Francisco, Department of Biology, San Francisco, California, USA

[u]Division of Pulmonary and Critical Care Medicine, Department of Internal Medicine, University of Michigan, Ann Arbor, Michigan, USA

[v]Social Science Program, University of Maine at Augusta, Augusta, Maine, USA

[w]University of Maine, Graduate School of Biomedical Science & Engineering, Bangor, Maine, USA

[x]Department of Biology, New Mexico State University, Las Cruces, New Mexico, USA

[y]Field of Soil and Crop Sciences, School of Integrative Plant Science, Cornell University, Ithaca, New York, USA

[z]National Institutes of Health, Clinical Center, Bethesda, Maryland, USA

[aa]School of Marine and Atmospheric Sciences, Stony Brook University, Stony Brook, New York, USA

[bb]Department of Psychology, University of Guelph, Guelph, Canada

[cc]Molecular Genetics and Microbiology, Duke University, Durham, North Carolina, USA

[dd]Eugene, Oregon, USA

[ee]Robert Larner College of Medicine, University of Vermont, Burlington, Vermont, USA

[ff]Finch Therapeutics, Somerville, Massachusetts, USA

[gg]Environmental Studies Program, University of Oregon, Eugene, Oregon, USA

[hh]Department of Philosophy, University of Oregon, Eugene, Oregon, USA

**Citation** Ishaq SL, Parada FJ, Wolf PG, Bonilla CY, Carney MA, Benezra A, Wissel E, Friedman M, DeAngelis KM, Robinson JM, Fahimipour AK, Manus MB, Grieneisen L, Dietz LG, Pathak A, Chauhan A, Kuthyar S, Stewart JD, Dasari MR, Nonnamaker E, Choudoir M, Horve PF, Zimmerman NB, Kozik AJ, Darling KW, Romero-Olivares AL, Hariharan J, Farmer N, Maki KA, Collier JL, O'Doherty KC, Letourneau J, Kline J, Moses PL, Morar N. 2021. Introducing the Microbes and Social Equity Working Group: considering the microbial components of social, environmental, and health justice. mSystems 6:e00471-21. https://doi.org/10.1128/mSystems.00471-21.

Address correspondence to Suzanne L. Ishaq, sue.ishaq@maine.edu.

Introducing the Microbes and Social Equity (MSE) Working Group, which connects microbiology with social equity research, education, policy, and practice to understand the interplay of microorganisms, individuals, societies, and ecosystems.

**ABSTRACT** Humans are inextricably linked to each other and our natural world, and microorganisms lie at the nexus of those interactions. Microorganisms form genetically

flexible, taxonomically diverse, and biochemically rich communities, i.e., microbiomes that are integral to the health and development of macroorganisms, societies, and ecosystems. Yet engagement with beneficial microbiomes is dictated by access to public resources, such as nutritious food, clean water and air, safe shelter, social interactions, and effective medicine. In this way, microbiomes have sociopolitical contexts that must be considered. The Microbes and Social Equity (MSE) Working Group connects microbiology with social equity research, education, policy, and practice to understand the interplay of microorganisms, individuals, societies, and ecosystems. Here, we outline opportunities for integrating microbiology and social equity work through broadening education and training; diversifying research topics, methods, and perspectives; and advocating for evidence-based public policy that supports sustainable, equitable, and microbial wealth for all.

**KEYWORDS** biopolitics, health disparities, social determinants of health, structural determinants of health, integrated research, microbiomes

## MICROCOSMS OF SOCIAL EQUITY

> **As an anthropologist,**…**what I love about the microbiome is that it brings together social intimacies of life with our biological selves in ways that show us that those two things are inextricably entangled.**
> —Amber Benezra, The Microbes and Social Equity spring 2021 seminar series, virtual, 10 March 2021, https://video.maine.edu/media/The+Global+MicrobiomeA+microbes+and+public+health+beyond+biology/1_nyje1v0b

Humans are inextricably linked to each other and our natural world through social, political, economic, and cultural interactions that have biological or ecological impacts. Microorganisms, including bacteria, archaea, fungi and other microbial eukaryotes (protists), and viruses, lie at the nexus of those relationships. Microbial transmission between individuals or from the environment is inevitable and ongoing; in our lifetimes, we encounter incredible microbial biodiversity, and in this way, microbes are the ties that bind us [1, 2].

Exposures to and long-term interactions with microorganisms are involved in the health, development, and security of individuals [3], societies [4–6], and ecosystems [7, 8]. For example, nourishing diets support microbial communities that benefit mental, physical, and immune system function(s) [9–11]. However, oppressive social structures result in financial, temporal, geographic, and logistical barriers to accessing healthy diets [12–14] and, thereby, beneficial microbiomes. Exposure to a variety of xenobiotics, including profligately employed antibiotics [15], is associated with depleted and distorted microbial communities [16–20]. Access to greenspace promotes microbial exposures [21–24] that benefit physical and mental health [25] but is impacted by a history of environmental racism [26]. Moreover, environmental microbiomes offer sustainable solutions to waste remediation, bioenergy production, food production, climate change mitigation, and, indeed, Earth's future [27–29], yet human activities fundamentally alter environmental microbial communities on which humans depend for our physical, social, and economic well-being [8, 30, 31].

Clean water, fresh air, healthy soils, nutritious foods, safe shelter, and the preservation of global biodiversity inherently depend on microbial communities [7, 32], and healthy human microbiomes depend on these resources. Access to these resources is a fundamental human right [33–35]; however, neither the resources nor the associated beneficial microbiomes are equitably distributed. Thus, the key to supporting healthy microbiomes (from guts to soils) is rooted in addressing social inequities (Table 1).

**Group Mission Statement.** The Microbes and Social Equity (MSE) Working Group posits that microbial exposures across ecosystems, urban and rural settings, and individuals are sociopolitical. Our purpose is to connect microbiology with social equity research, education, policy, and practice to understand the interplay of microorganisms, individuals, societies, and ecosystems. Collectively, we seek to generate and communicate

mSystems®

**TABLE 1** Glossary of terms

| Term | Definition |
| --- | --- |
| Biopolitics | The forms of political power concerned with the scientific management of biological processes and population dynamics of human, animal, and microbial life. Biopolitical resources are the processes managed through the scientific and administrative attempts to define, predict, or control human or non-human life, health, productivity, reproduction, and populations. See references 73 and 153. |
| Environmental justice | Equitable treatment and meaningful involvement of all people with respect to the development and implementation of environmental policies and practices. Equal access to environmental risks and benefits. See reference 145. |
| Epistemology | The theory of knowledge and the differentiation of belief from opinion. |
| Exposomics | The study of collective microbial and chemical exposures over time. See reference 100. |
| Intersectionality | The idea that identities, including, but not limited to, race, gender identity, ability, sexuality, and socioeconomic status, overlap in individuals, an acknowledgement that is required to bring social justice to the microbiome. In addition to referring to overlapping (nonadditive) identities, intersectionality refers to interlocking systems of oppression (e.g., racism, sexism, capitalism, ableism). See references 154, 155, and 162. |
| Microbial diversity | A diverse community contains more "types" of microorganisms and is analyzed by examining the number of types (i.e., species, strains, or functions) as well as their abundances and distributions in a host population or habitat. |
| Microbiome | The collection of microorganisms, which may include bacteria, archaea, fungi and other microbial eukaryotes (protists), and viruses, in a given habitat or host and their genomes, which are often used to characterize the collective organismal diversity and functional capacity of that community. See reference 147. |
| Neoliberalism | A set of political-economical and ideological principles and policies based on the view that every individual is an equal economic and social actor in a society best regulated by the "free" market. Its spread has depended upon a reconstitution of state powers such that privatization, finance, and market processes are emphasized. See reference 148. |
| Ontogeny | All physiological, developmental, and phenomenological events occurring during the processes underlying biological organization across an agent's lifetime. |
| Social equity | The concept that additional barriers exist for certain social groups that restrict access to public resources because of implicit or explicit biases and the active support of social policy, viewpoints, and public infrastructure that promote access to public resources in a way that dismantles these additional barriers. See reference 149. |
| Social determinants of health | The living, working, and local environmental conditions around a person that affect their health, their risk of harm, and health outcomes following medical interventions. See reference 150. |
| Social justice | The concept that wealth, economic opportunities, and financial privileges should be equitably distributed or accessible within a society. The practices of legal policy and law that facilitate more equitable distribution of economic opportunity when they are not. |
| Spatial justice | The concept that socially valued resources, such as natural and built environmental (i.e., infrastructure) resources, are not equitably distributed or accessible within a society. The practices of social and legal policy that facilitate equitable access to important resources. See reference 151. |
| Sustainability | The state in which a system is able to function with little-to-no additional outside inputs or with little-to-no waste. |
| Transdisciplinary research | Research that brings together different disciplinary perspectives to forge a new, synthetic field or framework, as opposed to interdisciplinary research, which brings together different disciplinary perspectives while keeping them distinct from one another. |

knowledge that will spark evidence-based public policy and practice, supporting equity and sustainability for all.

## MANY INDIVIDUAL CONTRIBUTIONS CREATED A COMMUNITY

What do microbes have to do with social equity? What began as a thought exercise in 2019 became an educational discourse (S. L. Ishaq's "Microbes and Social Equity" course materials) and an essay (1) and has since grown into an international community of like-minded researchers (The MSE Working Group and S. L. Ishaq). In line with the pivotal role of microbiomes and growing public attention to issues of social inequity and health outcome disparities, the MSE Working Group members represent diverse fields, e.g., anthropology, architecture, bioethics, bioinformatics, data science, ecology, engineering, genetics, medicine, microbiology, nutrition, psychology, and sociology, and exhibit expertise in various hosts, systems, and environments beyond human microbiomes. We are researchers, educators, practitioners, and policymakers spanning the globe and career levels. While the term "microbes and social equity" is novel, this concept has previously entered academic and public discourse (10, 36–43). We gratefully acknowledge these previous efforts to situate natural sciences within sociopolitical contexts.

**A Call to Collaboration.** Connecting diverse demographics, cultures, and geographic regions is an underappreciated opportunity in the microbiologist's domain. This work requires us to develop teams of biological and social scientists and policy experts who engage with affected communities to create solutions for substantial change. This creativity requires research journals to accommodate biological and anthropological data in the same piece, imaginative study endpoints that create meaningful change, and the intellectual freedom to weave detailed narratives when presenting research findings. For data-driven sciences, this collaboration requires interrogating one's own research within social contexts and existing biases (44). For example, previous work investigating microbial mechanisms of health disparities has focused on how environmental, structural, and racial politico-economic discrimination and other inequities influence microbiomes, instead of falsely assuming inherent biological differences between people of different races (38). Interventions that ignore social interactions or neglect the social determinants of health may fail to meet their goals (44, 45).

Links between social equity and microbial communities cannot be studied solely within the confines of the laboratory; community engagement and clear communication about research impact improve inclusivity (47, 48). At its worst, barriers to inclusivity exclude populations from health care and research, treating them as study objects (49) or as sources of extractable resources (50, 51), rather than equal partners. The disregard of cultural and personal dignity, as well as "othered" forms of knowledge (52, 53), creates a lasting atmosphere of institutional betrayal (54) in health care and research. We embark on this work with acknowledgment of colonial histories from which our universities and disciplines emerged (52, 55–57).

Microbiome science must consider applications that will benefit study populations in the foreseeable future, since direct, immediate intervention is unlikely at this stage of our understanding. Thus, we support an "ethics of care" (58) that requires microbiome researchers not only to attend to current predicaments of research participants, to support meaningful infrastructural changes, and to remain alert to possible commercial exploitation (38, 59–61) but also to consider that a participant's health extends well beyond the boundaries of "skin and skull," and to consider the microbiome as integral to the individual's inherent functioning (38, 46, 59–61). Certainly, this complicates the ways in which we understand our own physiology (44) and calls into question our self-conception (62, 63), our individuality (64), and, thus, some of the most central categories that we integrate into our ethical and political reasonings. Accordingly, collaboration within the MSE Working Group necessitates an explicit feminist, antiracist, anti-imperial, and anticolonial framework for knowledge production. Our goals are aspirational, and as with all abolitionist future-making projects which seek liberation from oppressive systems and institutions, our imperfect endeavor necessitates revision and reflexivity over time (65, 66).

## FOSTERING THE NEXT GENERATION OF RESEARCHERS

**In many global contexts, there is an epistemological divide between social and "other" sciences. MSE has the potential to offer a diverse—yet unifying— ground for transdisciplinary efforts.**
—Francisco J. Parada, MSE Working Group writing session, 2021

A central challenge in accelerating scientific progress on MSE issues is the need to train a generation of interdisciplinary scientists, many from minoritized groups (67), who can deploy complex systems thinking (68) to understand and model large, diverse, microbe-human systems (69). The goal is a novel transdisciplinary research and application agenda which combines disciplinary perspectives to forge a new, synthetic field or framework. In addition to domain-specific and mathematical approaches that drive the development of conceptual and quantitative theories, parallel training in "big data" analysis will be fundamental (70, 71). Lessons from other research programs indicate that future curricula will benefit from emphasizing both inductive and deductive scientific methods and the strategic combination of theory and data (72).

Further, integration between disciplines necessitates an epistemic change in the understanding of both the object of study and the observer (73). The success of transdisciplinary efforts requires novel funding and training opportunities as well as new epistemic frameworks and methods for integrating social, ethical, and justice issues into technoscientific practice and the design of technologies (74, 91, 156). Research initiatives, such as One Health (75, 76), which simultaneously considers human, social, and environmental health, or academic microbiome programs, such as the Oxford Interdisciplinary Microbiome Project (University of Oxford) and the Microbiome Initiative (University of California Riverside), foster interdisciplinary education and research to meld social and natural sciences. Similarly, "citizen science" initiatives engage both researchers and the general public. However, maintaining an interdisciplinary career or creating a transdisciplinary one requires governmental funding agencies and academic institutions to adopt creative or field-inclusive funding strategies (e.g., National Science Foundation program nsf19550) in order to promote a multifaceted scientific future (77–79).

**A Call to Integrated Curricula.** While grade school to graduate-level training requires an educational balance of biological science, social sciences, and humanities coursework, these disciplines are treated as epistemologically incompatible. Courses that meld these curricula (80; S. L. Ishaq's "Microbes and Social Equity" course materials) attract students from multiple disciplines and promote their agency in tackling seemingly intractable social problems through collaborative problem-solving assignments (1). Further, decades of attempts to diversify the science, technology, engineering, and mathematics (STEM) workforce have resulted in only modest increases (67). This necessitates a paradigm shift that centers and values marginalized voices in science curriculum development. Marginalized minorities (e.g., Indigenous peoples and nonwhite, sexual, and gender minorities) are usually left out of the system due to logistical (e.g., lack of infrastructure), technical (e.g., lack of formal education), and/or structural (e.g., lack of resources, institutional racism) circumstances. Centering intersections between microbiomes, human health, planetary health, social justice, and environmental justice signals to students the importance of an integrated worldview and is a means of promoting congruent scientific and cultural identities for those from marginalized groups (81, 82). Trainees have expressed a desire for integrated educational approaches (83) which ultimately promote the success of a diverse student population in the sciences (82, 84, 85).

Universities can actively support these efforts by (i) promoting and funding socially integrated science courses and programs; (ii) including and funding out-of-discipline students, educators, and researchers (86); (iii) collaborating with other institutions to implement multisite global classrooms; and (iv) changing the underlying model of education and research, which establishes a clear divide between basic and applied science (or as D. E. Stokes put it, rigor versus relevance [87]). Further, this assumes that the social benefits of research and education will eventually be derived, but this relationship is not linear. It is circular and complex; research and education embedded in people's social, cultural, and political realities mean inherently applying and creating knowledge. The modern conception of the educational system as a supermarket of knowledge deters itself from seamlessly applying that very same knowledge into broad understandings and meaningful interventions, programs, products, and devices. This is relevant because scientists are expert dichotomy creators; the paradigmatic example is the ages-long nature-versus-nurture debate, whose solution is almost never nature or nurture but both. Future scientific efforts—as an embedded dialogue between the world and the laboratory—should deal with the diverse realities afforded by rethinking the basic/applied dichotomy. Likewise, microbiology, a discipline that emerged from the study of microorganisms in contexts of hosts or ecosystems, has long spanned this divide given that Pasteur's seminal work makes both basic and applied contributions to science and society (87). The MSE Working Group naturally affords a setting for the simultaneous pursuit of both research and educational

programs with direct impacts on developing and testing novel hypotheses, as well as promoting and creating more effective practices.

## IDENTIFYING FUTURE RESEARCH

Identifying the most pressing research needs (37, 88, 89) in the emerging field of MSE is one of our fundamental aims. In late 2020, we invited researchers and practitioners from 18 institutions across the world to identify 20 important MSE-related research questions. To maximize rigor and diversity, the project involved a systematic approach adapted from previous methodology to identify research needs (89), and diverse backgrounds provided lateral thinking, fostered critical thinking (90), and enabled researchers to identify and refine novel and impactful agendas. In summary, individuals in research, education, clinical care, and policy are considering how microbiomes are a record of the material conditions and of the social relations to which a host has been exposed and how they impact human physical and mental health outcomes, food security and planning, or the environment writ large. The MSE Working Group expands our perception of microorganisms from solely biological entities to biopolitical resources and, consequently, serves as a target for intervention across social and ecological dimensions (157–161).

Inhabitants of prisons and homeless shelters and those who reside in neighborhoods zoned for industrial use can experience higher rates of exposure to pathogenic microorganisms (6, 92–94), strains of antibiotic-resistant bacteria (95), and industrial chemicals (26, 96, 97). These exposures can severely impact human health by altering the microbiome, causing illness and disease, or resulting in epigenetic changes (98, 99). The situated nature of microbial community dynamics, known as the exposome (100, 101), unveils complexities between environmental chemical and biological exposures with internal health consequences. With regard to their exposomes, some populations are disproportionately vulnerable. Novel hypotheses will use recent theoretical (102–104) and technological (105, 106) advancements to quantify the extent to which lifetime interconnections impact physiological and neurobehavioral dynamics and consequences for health, i.e., ontogeny. The challenge is manifold, as exposomics encounters (i) a scarcity of systematic multilevel data, (ii) an underdeveloped analytical framework to deal with those data, and (iii) the lack of an appropriate epistemological framework that would do justice (107) to the complexity of the phenomenon.

**A Call to Creative Design.** There are major logistical challenges to effectively recruiting and engaging diverse human study participants in microbiome projects (48, 108–110). Even when participants are compensated, there is a cost to participating in a research study that precludes true representation because it not only skews our understanding of microbiomes but also deepens the divide of research exclusionism. These costs include personal time (precluding those with multiple jobs, shift-based employment, or family care responsibilities), transportation to research facilities, and the time and mental energy needed to fit research activities into daily life (48, 111–113).

To overcome logistical challenges, creative techniques based on empathy and anthropological understanding are required to reduce these impacts (48, 110, 114, 115). These can include sample collection training (116); accounting for local resources, such as providing freezers or using alternate sample preservation methods, ensuring reliable power for laboratory equipment, and providing for transportation of samples from field sites; and/or using biological or environmental samples previously collected for another purpose (117). Establishing sustained and equitable partnerships with communities also requires an awareness of the barriers posed by the use of English as the primary language of science. While an expansion of scientific literacy will be critical to forming connections with many underserved communities, the process of literature review and knowledge generation would also benefit from expanding linguistic boundaries (118).

**A Call to Action.**

**To name a thing is to call it into existence, to give it agency. Today, we are named. Today, we become agents of change.**

—Suzanne L. Ishaq, MSE Working Group writing session, 2021

Even without an understanding of the effect of microorganisms on our lives, most

people recognize that individual health and well-being are common global goods (119). The benefits of considering social inequities in microbial ecology are both qualitative and quantitative, but for some audiences, "money talks." Fortunately, the benefits of social policies that promote equitable access to resources are strongly supported by demonstrable, repeatable returns on investment (120–127). Future research must examine the effects of neoliberal racial capitalism and its corollaries that promote market-based solutions, such as mass-produced commodities or lifestyles. For instance, what are the effects of industrial diets on human microbiomes or the impacts of political, economic, and social variables in shaping microbial communities (128, 129)?

We need comparative research that examines the structural constraints faced by people with microbe-associated disease (1), while staying vigilant about "magic bullet" biotechnological solutions that, while often endorsed by policymakers, rarely address the sociocultural, political, or economic factors that engender and reproduce health inequities on a global scale. Singling out nutritional deficiencies (e.g., golden rice to resolve vitamin A deficiency) without considering broader dietary contexts of food insecurity or placing an isolated focus on particular public health concerns (e.g., obesity interventions instead of food quality and access interventions) may lead to lopsided social outcomes which resolve health issues in those populations which had available resources for short-term treatment but do not supply sustainable, prevention-based solutions to underresourced populations (130, 131). Moreover, the neocolonial attitude of international aid and global/public health programs frequently equates "development" with Western hygiene and behaviors, with little regard for microbial community dynamics *in situ*. While microbiome sciences continue to identify certain biological patterns that are more reliably correlated with health and disease, it is still worth mentioning that "healthy microbiomes" are broadly situational and continue to defy standard definitions. We reiterate that "one microbiome does not fit all."

For a greater impact on public health and to achieve equitable health outcomes, policy efforts can focus on improving the local nutrition environment. While programs like the Supplemental Nutrition Assistance Program (SNAP) in the United States increase the food-purchasing power of lower-income households, increased food security may not translate into improved food quality (132), as neighborhoods of lower socioeconomic status have fewer supermarkets and more convenience, liquor, and fast-food outlets (133–135). Predatory marketing practices in these neighborhoods create higher proportions of advertisements promoting unhealthy foods (136, 137). It is crucial that policy efforts not only focus on providing enough calories but also use a combination of incentive (138, 139, 152) and disincentive (140, 141) programs to improve the affordability, availability, and accessibility of healthy options. Further, much remains unknown about the impact of food processing and food additives on microbial ecology, and we need microbiome research in partnership with the food industry and policy makers to mitigate potential harmful effects.

Healthy soils are intimately linked to human health through ecosystem processes like global nutrient cycling and food production, and more research should focus on how human-mediated land use change (i.e., deforestation, intensive agriculture, urbanization) impacts patterns of soil biodiversity (32, 142). For example, plant biodiversity, a primary driver of soil microbiome diversity and community structure, is lower in communities disenfranchised through "redlining" (inequitable development districting), demonstrating how systemic racism affects ecosystem biodiversity (143). Soil microbiomes offer promising metrics for assessing soil health across environmental change and through time (144). It is critical to understand how anthropogenic climate change impacts environmental microbes, as well as how microbes drive global change processes (8), and to consider this during environmental impact studies for infrastructure development or industry permitting to avoid ecosystem disruption.

By placing social justice as the central articulator between health and disease over a lifetime, the MSE Working Group provides a multidimensional perspective on social justice, situating the health consequences of exposure to depauperate or pathogenic

microbial communities among systemic issues, like disempowerment, threats of violence, abiotic environmental hazards, and others. Likewise, the MSE Working Group has a core focus on the effective translation of its principles into successful research outcomes and impactful public policy. Thus, the vision of the MSE Working Group necessitates engaging with public policy and pushing back on private interests that are both responsible for and seek to profit from microbial (and human) inequities. Finally, the MSE Working Group is poised to advance the concept of "microbial stewardship" (119) in fostering the resilience and sustainability of environmental microbial communities and promoting equity in the ways in which people are exposed to and interact with these communities. However, this cannot be accomplished through research alone. If we recognize microbiomes as a common good, then it requires all of us to progress our knowledge to action and advocate to protect that good.

## ACKNOWLEDGMENTS

We acknowledge the collective gifts of the lands and peoples which came before us and acknowledge that the institutions at which many of us work are located on lands which were taken from Indigenous peoples. We are grateful to the University of Oregon Robert D. Clark Honors College for hosting the original Microbes and Social Equity course taught by S. L. Ishaq in 2019, to the authors of the essay that resulted from that class, and to the University of Maine Institute of Medicine and the Established Program to Stimulate Competitive Research (EPSCoR) for materially and financially supporting the speaker series, virtual symposium, and other efforts of the MSE Working Group beginning in 2020. We are grateful to the additional current MSE Working Group members for their support and perspective on our general initiatives.

The MSE Working Group members who have contributed to this publication as consortium authors include Julian Damashek, Utica College, and Rachel Gregor, Massachusetts Institute of Technology.

We are grateful to the following funding agencies, who have supported the individuals in this group and their research efforts. S.L.I. is partially supported by the University of Maine through the Maine Agricultural and Forest Experiment Station (MAFES grant ME022102). F.J.P. is supported by the Agencia Nacional de Investigación y Desarrollo (ANID) through the Fondo Nacional de Desarrollo Científico y Tecnológico (FONDECYT) Iniciación en Investigación (program project no. 11180620 and regular project no. 1190610). P.G.W. is supported by a fellowship through the Cancer Education and Career Development Program (grant T32CA057699). E.W. is supported by a National Science Foundation Graduate Research Fellowship under grant 1937971. K.M.D. and M.C. are supported in part by the National Science Foundation Division of Environmental Biology under grant 1749206. A.K.F. is supported by the National Academies of Sciences, Engineering, and Mathematics National Research Council Associateship Program. A.J.K. is supported by an NIH F32 grant (no. 1F32HL150954-01). J.D.S. is supported by the Dutch Research Council (NWO/OCW) as part of the MiCRop Consortium program Harnessing the Second Genome of Plants (grant 024.004.014). N.F. is supported by intramural research funds from the National Institutes of Health Clinical Center. K.A.M. is supported by intramural research funds from the National Institutes of Health Clinical Center. J.L.C. is supported by the Gordon and Betty Moore Foundation's Experiment Model Systems (grant 4982). K.C.O. is supported through research funds from the Canadian Institutes of Health Research and The Social Sciences and Humanities Research Council of Canada.

The statements expressed in and contents of this article are those of the authors and do not reflect the official position of the National Institutes of Health, Department of Health and Human Services, or the U.S. Government.

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
