## [Reviewer comments · mSystems]

Introducing the Microbes and Social Equity Working Group: Considering the Microbial Components of Social, Environmental, and Health Justice

Suzanne Ishaq, Francisco Parada Flores, Patricia Wolf, Carla Bonilla, Megan Carney, Amber Benezra, Emily Wissel, Michael Friedman, Kristen DeAngelis, Jake Robinson, Ashkaan Fahimipour, Melissa Manus, Laura Grieneisen, Leslie Dietz, Ashvini Chauhan, Ashish Pathak, Sahana Kuthyar, Justin Stewart, Mauna Dasari, Emily Nonnamaker, Mallory Choudoir, Patrick Horve, Naupaka Zimmerman, Ariangela J. Kozik, Katherine Darling, Adriana Romero-Olivares, Janani Hariharan, Nicole Farmer, Katherine Maki, Jackie L Collier, Kieran O'Doherty, Jeffrey Letourneau, Jeff Kline, Peter Moses, and Nicolae Morar

Corresponding Author(s): Suzanne Ishaq, University of Maine

Review Timeline:

Submission Date:	April 15, 2021
Editorial Decision:	May 1, 2021
Revision Received:	May 11, 2021
Accepted:	May 11, 2021

Editor: Jack Gilbert

Reviewer(s): Disclosure of reviewer identity is with reference to reviewer comments included in decision letter(s). The following individuals involved in review of your submission have agreed to reveal their identity: Catherine Lozupone (Reviewer #2)

Transaction Report:

DOI: <https://doi.org/10.1128/mSystems.00471-21>

May 1, 2021

Dr. Suzanne Lynn Ishaq
University of Maine
Food and Agriculture
5763 Rogers Hall
Room 108
Orono, Maine 04469

Re: mSystems00471-21 (Introducing the Microbes and Social Equity Working Group: Considering the Microbial Components of Social, Environmental, and Health Justice)

Dear Dr. Suzanne Lynn Ishaq:

Thank you for submitting your manuscript to mSystems. We have completed our review and I am pleased to inform you that, in principle, we expect to accept it for publication in mSystems. However, acceptance will not be final until you have adequately addressed the reviewer comments.

Both reviewers were quite clear in their reviews, and while I may tend to disagree with some of the concerns about jargon, it might be pertinent to ensure that the language used is accessible to all peoples, as such some editing to achieve that wouldn't go amiss.

Thank you for the privilege of reviewing your work. Below you will find instructions from the mSystemseitorial office and comments generated during the review.

Preparing Revision Guidelines

For complete guidelines on revision requirements, please see the Instructions to Authors at [link to page]. **Submissions of a paper that does not conform to mSystems guidelines will delay acceptance of your manuscript.**

Due to the SARS-CoV-2 pandemic, our typical 60 day deadline for revisions will not be applied. I hope that you will be able to submit a revised manuscript soon, but want to reassure you that the

journal will be flexible in terms of timing, particularly if experimental revisions are needed. When you are ready to resubmit, please know that our staff and Editors are working remotely and handling submissions without delay. If you do not wish to modify the manuscript and prefer to submit it to another journal, please notify me of your decision immediately so that the manuscript may be formally withdrawn from consideration by mSystems.

Sincerely,

Jack Gilbert

Editor, mSystems

Journals Department
Reviewer comments:

Reviewer #1 (Comments for the Author):

This brief introduction/concept/opinion piece introducing the Microbes and Social Equity Working Group is interesting and generally well-written. It will undoubtedly spur some debate - but that is definitely a good thing. Parts of the article seem like a bit of a 'stretch' to me in trying to tie microbiome research into the ongoing, important discussions about social equity, but I think that is OK. Most of my comments detailed below are fairly minor and mostly focus on text where some clarification could be useful.

- I realize the authors are purposefully trying to be a bit controversial (which is OK!), but claiming that 'microbiomes are sociopolitical' (as done in the Abstract) seems a bit disingenuous and it is probably more accurate to say that that 'exposures' to microbiomes should often be studied in a sociopolitical context (as is clarified in the text). I only mention this as some readers may stop reading after they are told that microbiomes themselves are sociopolitical as the discussion that follows the abstract is actually far more nuanced and valuable than that simple slogan might suggest.

- What is an 'abolitionist future-making project'?

- What is the difference between 'transdisciplinary' and 'interdisciplinary'? (see paragraph 228-236 where the two terms seem to be used interchangeably). I realize this is explained in the glossary, but the introduction of this jargon seems to detract from the main message (in my opinion) and the nuances of these terms seems a bit unnecessary here.

- How are microbiomes a 'material record of social power relations'? This seems a bit jargon-y to me and I don't know what it really means.

- Lines 339-350 - the term 'neocolonial' is used twice here and I don't understand why 'magic bullet' type biotechnological solutions are effectively 'neocolonial'. Is an effective drug to treat malaria necessarily neocolonial (as just one example)?

- I realize that a glossary is provided, but I don't think a glossary should really be needed and it actually detracts from some of the points the authors are trying to make. Many of the points introduced could be made more strongly by avoiding some of this jargon (much of it derived from social science research) - recognizing that some of these specific terms may be unfamiliar to most of the readers of this article.

Reviewer #2 (Comments for the Author):

This paper introduces the Microbes and Social Equity (MSE) Working Group, which has the goal of connecting microbiology with social equity research, education, policy, and practice to understand the interplay of microorganisms, individuals, societies, and ecosystems. Broadly, the paper outlines opportunities for integrating microbiology and social equity work. Overall, the paper is thought-provoking and is highlighting an important topic. There are a few places though where I was unclear on the meaning of some statements, and these are detailed below.

1) I found the paragraph on lines 239-244 generally difficult to understand. I don't really understand what is meant by the second sentence "The success of interdisciplinary efforts requires novel opportunities that depend on future scientific agency to adopt this change." By "this change" do you mean change that research needs to understand the observer and not just object of study? What is "future scientific agency" referring to exactly? A brief description of what the "One Health" Research initiative is and how it "offers this training" exactly is needed. I don't know what this initiative is so hard to see how it supports the point. I am also unclear how a citizen science initiative offers training exactly and what that training even is in. What is an "inclusive funding strategy" exactly - I understand the point that offering "transdisciplinary" funding would be important for interdisciplinary research but not sure what is meant by inclusive funding in this context. Can any specific examples of existing transdisciplinary and or inclusive funding opportunities be brought up as an example of what we need more of?

2) Lines 268-270. "Further, this assumes that the social benefits of research and education will eventually be derived, but this relationship is not linear. Microbiology, a discipline that emerged from the study of microorganisms in contexts of hosts or ecosystems, has long spanned this divide (91)." I am finding hard to follow here. If it is not linear what is it and why does it matter? How has Microbiology "spanned this divide" exactly?

3) What is "neoliberal racial capitalism and its corollaries" (line 335) or "neocolonial 'magic bullet' type biotechnological solutions" (line 340) - seems very jargony.

4) Lines 343-346 - I think the points being made here need to be spelled out a little better. Not sure how the examples given of Golden rice for vitamin A deficiency or obesity interventions are leading

to "lopsided outcomes." Golden rice is adding a deficient nutrient to a staple crop that local farmers can grow so seems like a good initiative. Why would an obesity intervention be a bad approach? What is meant by a "lopsided outcome" exactly? Can you be more specific?

Reviewer comments:

Reviewer #1 (Comments for the Author):

This brief introduction/concept/opinion piece introducing the Microbes and Social Equity Working Group is interesting and generally well-written. It will undoubtedly spur some debate - but that is definitely a good thing. Parts of the article seem like a bit of a 'stretch' to me in trying to tie microbiome research into the ongoing, important discussions about social equity, but I think that is OK. Most of my comments detailed below are fairly minor and mostly focus on text where some clarification could be useful.

- **Response: We are grateful to the reviewers for their careful reading of our manuscript and thoughtful comments, and in particular, for the suggestions where we can make our text more accessible. As you might imagine, with so many authors from so many fields, we had quite a bit of jargon to learn from each other, and we would like to offer our readers the same opportunity. We have revised our text to clarify our meaning as suggested.**

- I realize the authors are purposefully trying to be a bit controversial (which is OK!), but claiming that 'microbiomes are sociopolitical' (as done in the Abstract) seems a bit disingenuous and it is probably more accurate to say that that 'exposures' to microbiomes should often be studied in a sociopolitical context (as is clarified in the text). I only mention this as some readers may stop reading after they are told that microbiomes themselves are sociopolitical as the discussion that follows the abstract is actually far more nuanced and valuable than that simple slogan might suggest.

- **Response: We appreciate this perspective, and we had originally struggled to grab reader attention and explain ourselves to a general audience within the Abstract word limit. We have revised this line to read: “In this way, *microbiomes have sociopolitical contexts that must be considered.*”**

- What is an 'abolitionist future-making project'?

- **Response: It means moving toward liberation of all from systems and institutions of oppression and domination, in other words, a truly equitable society. In the interest of word count, we have provided this additional language to make this idea more clear for readers unfamiliar with the rich literature on abolition.**

- What is the difference between 'transdisciplinary' and 'interdisciplinary'? (see paragraph 228-236 where the two terms seem to be used interchangeably). I realize this is explained in the glossary, but the introduction of this jargon seems to detract from the main message (in my opinion) and the nuances of these terms seems a bit unnecessary here.

- **Response:** We have revised our use of these terms in this paragraph to remove the ambiguity. As such, the initial sentence now reads as: “The goal is a novel *transdisciplinary* research and applications agenda which combines disciplinary perspectives to forge a new, synthetic field or framework.” In the following sentences, we removed the words “interdisciplinary” where it was used superfluously, and revised one instance to “transdisciplinary” as we were trying to comment on how funding agencies will need to rethink funding calls to support these new research frameworks.

- How are microbiomes a 'material record of social power relations'? This seems a bit jargon-y to me and I don't know what it really means.

- **Response:** For the sake of clarity, we have edited this sentence the following way: “microbiomes are a record of the material conditions and of the social relations that the host has been exposed to”.

The assumption is not that microbiomes function as a passive blank slate so that material conditions and social relations merely leave a print on them. Our assumption is that, given the dynamic nature of microbiomes at the interface of host and larger-scales environmental conditions, microbiomes seem to rapidly respond to material conditions and to social relations and to carry with them a trace/ record of that response. For example, David and colleagues (2014) have documented how diet rapidly alters the human gut microbiome. It’s amazing to see that in the case of ‘Subject A Gut’, one can specifically notice the impact on lifestyle, in this case the impact of living abroad, on one’s gut microbial community. Similarly, researchers have also looked at the impact of social relations on one’s microbiome (e.g. dogs or other humans) and they have noticed that gut and skin microbiomes of humans are more similar to those of cohabiting humans and dogs (Song & el. 2013; Brito & al. 2019). This is the way in which we believe that microbiomes are a record of the material conditions and of the social relations that the host has been exposed to.

- Brito, I. & al. 2019. “Transmission of human-associated microbiota along family and social networks”, *Nature Microbiology*, 4:964-971.
- David, L. & al. 2014. “Host lifestyle affects human microbiota on daily timescales”, *Genome Biology*, 15, R89, <https://doi.org/10.1186/gb-2014-15-7-r89>
- Song, S.J. & al. 2013. “Cohabiting family members share microbiota with one another and with their dogs”, *eLife*, 2:e00458, DOI: 10.7554/eLife.00458

- Lines 339-350 - the term 'neocolonial' is used twice here and I don't understand why 'magic bullet' type biotechnological solutions are effectively 'neocolonial'. Is an effective drug to treat malaria necessarily neocolonial (as just one example)?

- **Response: We edited the text so that the term ‘neocolonial’ only appears once. To further clarify the problematic aspects of magic bullet solutions, we added additional text to explain how these approaches often fail to address the underlying structural factors that reproduce health inequities. This better addresses the above question about a malaria drug-- while new technology of course saves lives, we are concerned with the frequent lack of attention paid to systemic inequities that dictate how resources are administered across populations. We also moved the subsequent sentence, where ‘neocolonial’ appeared for the second time, to further down in the paragraph, as it more clearly connects to the final sentence of the paragraph.**

- I realize that a glossary is provided, but I don't think a glossary should really be needed and it actually detracts from some of the points the authors are trying to make. Many of the points introduced could be made more strongly by avoiding some of this jargon (much of it derived from social science research) - recognizing that some of these specific terms may be unfamiliar to most of the readers of this article.

- **Response: We appreciate this perspective, and have had similar discussions within the group about the need for a glossary. In the end, we decided that introducing jargon from multiple fields was necessary to teach this terminology to new audiences, and, to reach those new audiences: to show up in online search results. We have tried to address the accessibility of our language by revising our text and clarifying our meaning around some of these terms where we use them- we previously had more of this but had to remove quite a bit of text to even come near the recommended word count. In the end, we felt that the glossary helped make this piece more approachable and decided to keep it; even though we expect that most mSystems readers will be from the microbiome fields, we hope to use this piece to teach these concepts to new and broader audiences.**

Reviewer #2 (Comments for the Author):

This paper introduces the Microbes and Social Equity (MSE) Working Group, which has the goal of connecting microbiology with social equity research, education, policy, and practice to understand the interplay of microorganisms, individuals, societies, and ecosystems. Broadly, the paper outlines opportunities for integrating microbiology and social equity work. Overall, the paper is thought-provoking and is highlighting an important topic. There are a few places though where I was unclear on the meaning of some statements, and these are detailed below.

- **Response: We are grateful to the reviewers for their careful reading of our manuscript and thoughtful comments, and in particular, for the suggestions where we can make our text more accessible. As you might imagine, with so many authors from so many fields, we had quite a bit of jargon to learn from each other, and we would like to offer our readers the same opportunity. We have revised our text to clarify our meaning as suggested.**

1) I found the paragraph on lines 239-244 generally difficult to understand. I don't really understand what is meant by the second sentence "The success of interdisciplinary efforts requires novel opportunities that depend on future scientific agency to adopt this change." By "this change" do you mean change that research needs to understand the observer and not just object of study? What is "future scientific agency" referring to exactly?

- **Response: We have revised this sentence to reflect that we mean ‘agency’ as in ability to enact changes: “The success of interdisciplinary efforts requires novel funding and framework opportunities that depend on the intellectual freedom/scientific *agency* of researchers to adopt this epistemic change (76).”**

A brief description of what the "One Health" Research initiative is and how it "offers this training" exactly is needed. I don't know what this initiative is so hard to see how it supports the point.

- **Response: We have revised this sentence and added more description: “Research initiatives such as One Health (77, 78) which simultaneously considers human, social, and environmental health, or academic microbiome programs (79, 80), foster interdisciplinary education and research to meld social and natural sciences.”**

I am also unclear how a citizen science initiative offers training exactly and what that training even is in.

- **Response: We have revised this sentence and added more description: “Similarly, ‘citizen science’ initiatives engage both researchers and the general public.”**

What is an "inclusive funding strategy" exactly - I understand the point that offering "transdisciplinary" funding would be important for interdisciplinary research but not sure what is meant by inclusive funding in this context. Can any specific examples of existing transdisciplinary and or inclusive funding opportunities be brought up as an example of what we need more of?

- **Response: We have revised this to “creative or field-inclusive funding strategies (e.g. National Science Foundation nsf19550)”**

2) Lines 268-270. "Further, this assumes that the social benefits of research and education will eventually be derived, but this relationship is not linear. Microbiology, a discipline that emerged

from the study of microorganisms in contexts of hosts or ecosystems, has long spanned this divide (91)." I am finding hard to follow here. If it is not linear what is it and why does it matter? How has Microbiology "spanned this divide" exactly?

- **Response: We apologize for the opacity of our previous statement, we had removed much of the original description prior to our submission. We have clarified our point: “Further, this assumes that the social benefits of research and education *will eventually be derived*, but this relationship is not linear. It is circular and complex; research and education embedded in people’s social, cultural, and political realities means inherently applying *and* creating knowledge. The modern conception of the educational system as a *supermarket of knowledge* deters itself from seamlessly applying that very same knowledge into broad understandings and meaningful interventions, programs, products and devices. This is relevant because scientists are expert dichotomy creators: the paradigmatic example is the ages-long *nature versus nurture* debate, whose solution is almost never nature *or* nurture, but *both*. Future scientific efforts -as an embedded dialogue between the world and the laboratory- should deal with the diversity realities afford by re-thinking the basic/applied dichotomy. Likewise, microbiology, a discipline that emerged from the study of microorganisms in contexts of hosts or ecosystems, has long spanned this divide given that Pasteur’s seminal work makes *both basic and applied* contributions to science and society (91).”**

3) What is "neoliberal racial capitalism and its corollaries" (line 335) or "neocolonial 'magic bullet' type biotechnological solutions" (line 340) - seems very jargony.

- **Response: These are widely used terms in the social sciences and humanities but we recognize that they warrant slightly more definition in the context of this journal as well as for advancing our interdisciplinary work. We have added more explanation of these terms in the text: “Future research must examine the effects of neoliberal racial capitalism and its corollaries which promote market-based solutions such as mass-produced commodities or lifestyles. For instance, what are the effects of industrial diets on human microbiomes, or the impacts of political, economic, and social variables in shaping microbial communities (132, 133)?”**

4) Lines 343-346 - I think the points being made here need to be spelled out a little better. Not sure how the examples given of Golden rice for vitamin A deficiency or obesity interventions are leading to "lopsided outcomes." Golden rice is adding a deficient nutrient to a staple crop that local farmers can grow so seems like a good initiative. Why would an obesity intervention be a bad approach? What is meant by a "lopsided outcome" exactly? Can you be more specific?

- **Response: We have rephrased these to clarify our meaning and more explicitly describe our examples: “We need comparative research that examines the structural constraints faced by people with microbe-associated disease (1), while staying vigilant of “magic bullet” biotechnological solutions that, while often endorsed by policymakers, rarely address the sociocultural, political, or economic factors that engender and reproduce health inequities on a global scale. Singling out nutritional deficiencies (e.g., Golden Rice to resolve vitamin A deficiency) without considering broader dietary contexts of food insecurity , or placing an isolated focus on particular public health concerns (e.g., obesity interventions instead of food quality and access interventions), may lead to lopsided social outcomes which resolve health issues in those populations which had available resources for short-term treatment, but do not supply sustainable, prevention-based solutions to under resourced populations (134, 135). Moreover, the neocolonial attitude of international aid and global/public health programs frequently equates "development" with Western hygiene and behaviors, with little regard for microbial community dynamics in situ.”**

May 11, 2021

Dr. Suzanne Lynn Ishaq
University of Maine
Food and Agriculture
5763 Rogers Hall
Room 108
Orono, Maine 04469

Re: mSystems00471-21R1 (Introducing the Microbes and Social Equity Working Group: Considering the Microbial Components of Social, Environmental, and Health Justice)

Dear Dr. Suzanne Lynn Ishaq:

Thank you for your careful response and edits.

Your manuscript has been accepted, and I am forwarding it to the ASM Journals Department for publication. For your reference, ASM Journals' address is given below. Before it can be scheduled for publication, your manuscript will be checked by the mSystems senior production editor, Ellie Ghatineh, to make sure that all elements meet the technical requirements for publication. She will contact you if anything needs to be revised before copyediting and production can begin. Otherwise, you will be notified when your proofs are ready to be viewed.

- Minimum resolution of 1280 x 720
- .mov or .mp4. video format
- Provide video in the highest quality possible, but do not exceed 1080p
- Provide a still/profile picture that is 640 (w) x 720 (h) max

We recognize that the video files can become quite large, and so to avoid quality loss ASM

suggests sending the video file via <https://www.wetransfer.com/>. When you have a final version of the video and the still ready to share, please send it to Ellie Ghatineh at eghatineh@asmusa.org.

Sincerely,

Jack Gilbert
Editor, mSystems

Journals Department
Phone: 1-202-942-9338